# Mark–Release–Recapture Trial with *Aedes albopictus* (Diptera, Culicidae) Irradiated Males: Population Parameters and Climatic Factors

**DOI:** 10.3390/insects15090685

**Published:** 2024-09-11

**Authors:** Fátima Isabel Falcão Amaro, Patricia Soares, Enkelejda Velo, Danilo Oliveira Carvalho, Maylen Gomez, Fabrizio Balestrino, Arianna Puggioli, Romeo Bellini, Hugo Costa Osório

**Affiliations:** 1Centre for Vectors and Infectious Diseases Research Doutor Francisco Cambournac (CEVDI), National Institute of Health Doutor Ricardo Jorge (INSA), Avenida da Liberdade 5, 2965-575 Palmela, Portugal; fatima.amaro@insa.min-saude.pt (F.I.F.A.); patseraos@gmail.com (P.S.); 2Environmental Health Institute (ISAMB), Faculty of Medicine, University of Lisbon, Av. Prof. Egas Moniz, Ed. Egas Moniz, Piso 0, Ala C, 1649-028 Lisboa, Portugal; 3NOVA National School of Public Health, Public Health Research Centre, Comprehensive Health Research Center, NOVA University Lisbon, 1070-312 Lisbon, Portugal; 4Centre of Statistics and its Applications (CEAUL), Faculty of Sciences, University of Lisbon, 1749-016 Lisbon, Portugal; 5Department of Epidemiology and Control of Infectious Diseases, Institute of Public Health, Tirana 1001, Albania; dikollikela@gmail.com; 6Insect Pest Control Subprogramme, Department of Nuclear Sciences and Applications, Joint FAO/IAEA Centre of Nuclear Techniques in Food and Agriculture, International Atomic Energy Agency, 1400 Vienna, Austria; deoliveiradanilo@ufl.edu (D.O.C.); m.gomez-pacheco@iaea.org (M.G.); 7Department of Entomology and Nematology, University of Florida, Gainesville, FL 32611, USA; 8Centro Agricoltura Ambiente “G. Nicoli”, Department of Sanitary Entomology and Zoology, 40014 Crevalcore, Italy; fabriziobalestrino@hotmail.com (F.B.); apuggioli@caa.it (A.P.); rbellini@caa.it (R.B.)

**Keywords:** *Aedes albopictus*, mark–release–recapture, mosquitoes, Portugal, sterile insect technique, vector control

## Abstract

**Simple Summary:**

*Aedes albopictus* mosquitoes spread arboviruses like dengue, Zika, or chikungunya. The Sterile Insect Technique (SIT) can be used as a prevention and control tool against *Ae. albopictus* populations. Mark–release–recapture (MRR) trials are fundamental to estimate the size of the wild population in target areas and to determine the dispersal capacity and survival of sterile males in open field conditions. Environmental conditions can affect the MRR trial’s results; thus, the influence of climatic factors on the first MRR trial with *Ae. albopictus* marked sterile males conducted in Portugal was analyzed. In October 2022, for three consecutive weeks at two different release points, 84,000 sterile males were released over 50 ha of the study area, in the municipality of Faro, Southern Portugal, and mosquitoes were recaptured by human landing collection (HLC) one, two, four, and six days after release. Distance travelled, daily survival, and life expectancy in the field were estimated for the released sterile males and the influence of climatic factors was analyzed. Despite no statistically significant association being found with humidity, temperature, and precipitation, it is crucial to plan MRRs, considering weather conditions for a more efficient application of the SIT in an integrated vector management program.

**Abstract:**

*Aedes albopictus* is considered one of the major invasive species in the world and can transmit viruses such as dengue, Zika, or chikungunya. The Sterile Insect Technique (SIT) can be used to suppress the native populations of *Ae. albopictus*. Mark–release–recapture (MRR) studies are crucial to support the development of the release strategy during the SIT application. Meanwhile, weather conditions can affect the MRR trial’s results and it is critical to understand the influence of climatic factors on the results. In October 2022, 84,000 irradiated sterile males were released for three consecutive weeks in Faro, Southern Portugal. Mosquitoes were recaptured by human landing collection (HLC) one, two, four, and six days after release. Generalized linear models with a negative binomial family and log function were used to estimate the factors associated with the number of recaptured mosquitoes, prevalence ratios, and the 95% confidence intervals (CIs). A total of 84,000 sterile male mosquitoes were released, with 528 recaptured (0.8%) by HLC. The prevalence of recaptured mosquitoes was 23% lower when the wind intensity was moderate. Marked sterile males had an average median distance travelled of 88.7 m. The median probability of daily survival and the average life expectancy were 61.6% and 2.1 days, respectively. The wild male population estimate was 443.33 males/ha. Despite no statistically significant association being found with humidity, temperature, and precipitation, it is important to consider weather conditions during MRR trial analyses to obtain the best determinant estimation and a more efficient application of the SIT in an integrated vector management program.

## 1. Introduction

Vector-borne diseases (VBDs) greatly impact human health, accounting for more than 17% of all infectious diseases worldwide [1,2]. The mosquito *Aedes albopictus* (Skuse, 1884), also known as the Asian Tiger Mosquito, is a competent vector species of a wide range of viruses and nematodes of the genus *Dirofilaria* spp., which raises a big concern for veterinary and public health [3]. This mosquito species can transmit arboviruses, causing diseases such as dengue, Zika, yellow fever, Japanese encephalitis, and chikungunya [4]. According to the Global Invasive Species Database, *Ae. albopictus* is the most invasive mosquito species in the world, and it has become a significant pest of public health concern in many European countries [5].

Autochthonous transmission of dengue and chikungunya viruses related to *Ae. albopictus* has been recently reported in Europe. Chikungunya outbreaks occurred in France in 2010, 2014, and 2017, as well as in Italy in 2007 and 2017 [6,7,8,9,10]. From 2010 until the present time, dengue autochthonous cases have also occurred in Croatia (2010), France (2010, 2013–2015, 2018–2023), Spain (2018, 2019, 2022, 2023), and Italy (2020, 2023) [11]. France presented Europe’s largest documented transmission event, with 65 confirmed cases [12].

Portugal has consistently been identified as a high-risk region for the introduction and establishment of *Ae. albopictus*. The first documented introduction was in 2017, in the municipalities of Penafiel (northern region) and Loulé (southern region), and both occurred as separated events [13,14]. These introductions were reported under the frame of the National Vector Surveillance Network (REVIVE), coordinated by the Ministry of Health’s National Institute of Health (INSA) and the Centre for Research on Vectors and Infectious Diseases (CEVDI). Since these findings, the REVIVE has reported the annual seasonal activity of *Ae. albopictus* [15].

Mosquito control is still essentially based on the widespread use of insecticides, despite the increasing concerns about their impact on other living organisms and ecosystems; the main problem of their use is the selection of naturally resistant mosquito populations, with the increased use of insecticides [16]. In the absence of efficient control methods, this invasive mosquito species might easily disseminate and establish in other Portuguese regions, thus increasing the risk of mosquito-borne disease outbreaks on a larger scale. Based on the present scenario in which our options to control mosquitos are limited, the World Health Organisation (WHO) expressed the urgent need for alternative mosquito control methods [16].

At this stage, and aiming to mitigate the potential impact of *Ae. albopictus* in transmitting pathogens causing human diseases in the country, Portugal is building capacities for evaluating the Sterile Insect Technique (SIT) as a sustainable tool to suppress *Ae. albopictus* populations at the national territory level using an area-wide integrated pest management (AW-IPM) approach. For this purpose, Portugal has undertaken an SIT pilot project in Gambelas, a neighborhood located in the Faro municipality, southern region of the Algarve. The project was implemented by the CEVDI/INSA with the technical support of the International Atomic Energy Agency (IAEA) technical cooperation programme (IAEA TC POR5006—Integrating the Sterile Insect Technique (SIT) in the Control of the Invasive Vector Mosquito *Aedes albopictus* in Portugal).

The SIT is an environmentally friendly insect pest control method based on the mass rearing and sterilization, using ionizing radiation, of males from the target species. This is followed by systematic, area-wide, continuous releases over defined areas, where they mate with wild females, compromising the offspring, resulting in a declining pest population over the generations since it is a species-specific technique [17,18]. The sterile males are not self-reproductive and do not introduce non-native species into an ecosystem. Thus, they cannot establish in the environment and change the local genetic diversity. No target genetic modification (or gene editing) is involved in the sterilization mechanism [16,19]. The SIT has been successfully used to suppress relevant insect pests, such as *Cochliomyia hominivorax* in North and Central America [20] and the eradication of *Ceratitis capitata* in countries of Central and South America [21]. However, the application of the SIT for mosquitoes, especially for *Ae. albopictus*, is in the early stages and several field trials are underway to assess and optimize the SIT package to allow the technology to become more comprehensive and operationally efficient in reducing mosquito populations [19]. SIT field trials in Italy and Germany showed a reduction in their local *Ae. albopictus* populations, achieving 70–80% and 84% egg sterility, respectively, in defined target areas [22,23].

Testing and implementing the SIT for vector control is challenging; therefore, it is recommended to follow a phased conditional approach (PCA), meaning that several activities and milestones must be accomplished before scaling up the technology from the pilot to an operational level for mosquito control [18]. The preparatory period is the baseline data collection phase, which will determine several fundamental parameters to guarantee success before adventuring and releasing mosquitoes in the field. Part of it includes mark–release–recapture (MRR) studies, which provide valuable information on the ability of the irradiated sterile males to survive and disperse in the open environment, and also estimate the wild population abundance under field conditions. To achieve this goal, a set of MRR studies were conducted to (i) estimate the size of the wild population of *Ae. albopictus* in the target area, (ii) determine the dispersal capacity and survival of sterile males in open field conditions, and (iii) understand the influence of climatic factors on the recapture of mosquitoes, which could affect the effectiveness of MRR and SIT trials. These steps are essential for the subsequent routine release of sterile males during the suppression phase of the pilot project.

## 2. Materials and Methods

### 2.1. Study Area

The mark–release–recapture trials were conducted in Gambelas, a small neighborhood with an area of about 50 ha, located in the Algarve region, municipality of Faro, Montenegro parish, which comprises an area of 23.24 km^2^ and has 8613 inhabitants (PORDATA, census 2021). The Gambelas neighborhood is close to the University of Faro, relatively well isolated by open fields and pine forests, and is less than five kilometers from the Faro airport, where a weather station is set up (Figure 1). The target area is characterised by an urban setting consisting of houses with private gardens offering suitable ecological conditions for *Ae. albopictus* reproduction due to continuous vegetation representing resting places, breeding site availability, and hosts for blood meals. No ultra-low-volume or residual application of any insecticide was performed in the target area. The climatic conditions are characterised by a warm and temperate atmosphere, with much more rainfall in winter than in summer. According to Köppen and Geiger, the climate classification is a hot-summer Mediterranean climate (Csa). The average temperature is 18.0 °C and annual precipitation is approximately 499 mm.

### 2.2. Mark–Release–Recapture Study

The mark–release–recapture trials were conducted from 11 to 31 October 2022, when the *Ae. albopictus* annual seasonal activity is highest, according to baseline data [24]. A monitoring network based on 20 ovitraps was set in the target area since April 2022. Two release points were selected and 40 human landing collection (HLC) georeferenced stations were set outdoors on the seven sequential annuli outlined (Figure 1), each with increments of 50 m up to 350 m, resulting in 27 ha of covered area combining both central points.

### 2.3. Mosquito Production

The *Ae. albopictus* sterile males used for the study were produced at the BSL2 facility of Centro Agricoltura Ambiente (CAA), in Crevalcore, Italy. The mosquito colony originated from eggs collected in the field in Faro through ovitraps during the baseline data collection phase. A few thousand eggs of the Portuguese strain were then shipped to CAA for mass rearing under controlled conditions (28 ± 1 °C, 80 ± 5% RH, 14:10 L:D) using mass-rearing cages for adults and larval-rearing units consisting of a mechanized stainless-steel rack holding 50 large trays [25,26].

Pupae were size-based sexed and males were collected by means of an automatic sex sorter and counted using a volumetric cylinder. A sample of about 300 individuals was checked to evaluate that the percentage of residual females was lower than 1%.

Male pupae aged 2–48 h were irradiated in water at a dose of 55 Gy, previously defined to ensure about 99% sterility, using the X-ray irradiator Radgil2 (Gilardoni, Milan, Italy).

For the long-distance transport, sterile male adults were immobilized in a cold room at 8 °C for 15 min, marked with fluorescent powder following FAO/IAEA guidelines [27], and packed in stacked polypropylene plastic cups (350 mL, 11.5 cm in diameter, 5 cm in height) by placing around 2000 adults per cup. The temperature inside the package was maintained around 10–12 °C using phase-change-material packs (PCM-ClimSel^TM^) conditioned at the proper temperature [23,27,28,29].

The sterile males were delivered to the field laboratory by DHL flight express service and the time span between leaving the production facility and field release was around 28 h.

### 2.4. Mosquito Release Protocol

Three releases were performed at two different release points (with a distance of approximately 50 m between them) at one-week intervals during three consecutive weeks. Sterile mosquitoes were released in the afternoon between 4–5 pm from the ground, from two single-points, both located in the central zone of the study site and spaced with approximately 50 m between them (Figure 1). Mosquito batches arrived at the field laboratory between 2–3 pm and were immediately transported to the release site. Mosquitoes marked with different fluorescent powder colors at CAA were released from one of the single release points. For the field releases, the release containers were placed directly over a white cloth on the ground, the lids were removed and sterile male mosquitoes were allowed to fly out for 60 min. Afterwards, all mosquitoes that did not fly out were taken back to the laboratory to estimate the effective number of released mosquitoes.

### 2.5. Mosquito Recapture Protocol

Field samples were performed by human landing collection (HLC). Forty HLC stations were set up following a radial pattern from both central release points, as represented in Figure 1. The HLC stations were deployed over seven concentric annuli, spaced 50 m apart and with a maximum station distance of 350 m. For the HLC, the technicians used as an attractant a black bag of 50 L covering an empty box, exposed their legs and used electric aspirators to collect mosquitoes during landing. They were then transferred to containers for further identification and counting. HLCs began one day after release and were performed at days two, four, and six after the release during three consecutive weeks. The HLC stations were grouped into three clusters to facilitate the technicians to move similar distances within the defined collection period (i.e., five minutes per HLC station). The collected field samples were then transported to the field laboratory for the count record, and species and sex identification. All *Ae. albopictus* specimens collected from HLCs were carefully analyzed under ultraviolet light to detect the fluorescent dust and marking status.

### 2.6. Population Parameters

#### 2.6.1. Dispersal Capacity

Distances from the release point and the HLC stations were defined using Vincenty’s formula for geodesic distance with an accurate ellipsoidal model of the Earth [30,31]. We divided the study area into seven circular concentric annuli, at 50 m intervals from the release sites (Figure 1), to estimate the mean distance travelled (MDT) and the flight range (FR) for each batch of released marked sterile mosquitos. We previously calculated the estimated recapture (ER) and the median distance between the annulus to calculate the MDT [32,33].
$${ER}_{c,a}=\;\frac{{Recapture}_{c,a}}{{Traps}_{c,a}}\;\times \;\left(\frac{{Area}_{a}}{Total\;area}\;\times \;Total\;traps\right)$$
where *c* corresponds to the color of the marked mosquitoes and *a* to the annulus.
$${MDT}_{c}=\;\sum \left(\frac{{ER}_{c,a\;}\times \;{mean\_distance}_{a}}{sum({ER}_{c})}\right)$$

We used linear regression to estimate the FR. Our outcome was the logarithm of the median distance for each annulus, and our independent variable was the cumulative sum of the ER. To obtain the 50% and 90% FR, we predicted 50% and 90% of the maximum cumulative sum of the ER, respectively.

#### 2.6.2. Wild Population Estimation

For estimating the wild male population size, we used the Fisher–Ford index. Only the first recapture day for each released marked batch was considered for the wild population estimation. We calculated estimates for each color and used these estimates to bootstrap an estimate for the population with 95% bias-corrected accelerated confidence intervals. Fisher–Ford estimates were obtained considering the estimated probability of daily survival (PDS) [estimation of this parameter is explained in the next subsection], releases, the number of captured males, and recaptures.
$${Fisher\;Ford}_{c}=\;\frac{{PDS}_{c}*capturedM*Releases}{{recapture}_{c}}-\;\left(Releases*{PDS}_{c}\right)$$
where *c* corresponds to the color, and *captured M* to the number of captured males—wild and sterile.

#### 2.6.3. Daily Survival

We considered a linear model to estimate the PDS. Our outcome was the logarithm of recaptured marked mosquitoes, plus one to account for days without recapture, and our independent variable was the number of days since release. Average life expectancy (ALE) was defined as one divided by minus the logarithm of the PDS. We bootstrapped the different PDS estimates to get an overall PDS estimate for the sterile population and estimated a 95% bias-corrected accelerated confidence interval (95% BCa CI).

### 2.7. Weather Data

Weather data were extracted from the weather station at Faro, under the Portuguese Institute of the Sea and the Atmosphere (IPMA, https://www.ipma.pt/pt/index.html; https://api.ipma.pt/ accessed on 17 February 2023), which collects data on climatic conditions daily and hourly. We considered temperature, humidity, precipitation, wind direction, and intensity, categorized as weak (<15 km/h) and moderate (15–35 km/h). To consider weather data at the time of release, if they occurred within the first 30 min of the hour, we rounded the time down to the current hour, and if it happened beyond 30 min, we rounded up to the next hour. For example, at h:00 to h:29, then h = h; when h:30 to h:59, then h = h + 1. We rounded the time to merge with data from the weather station at Faro, which are collected hourly.

### 2.8. Data Analysis

We used a multinomial model to assess differences between mosquitos marked with different colors. The independent variables were the distance and the number of captured mosquitoes. We also used a multinomial model to assess climatic differences between the three weekly rounds of releases. We extracted data from the weather station at Faro, using the same rationale mentioned in the Section 2.4 (MRR protocol). Temperature and humidity were centered on the mean to facilitate interpretation. We used generalized linear models with a negative binomial family and log function to estimate factors associated with the number of captured mosquitoes. First, we considered non-climatic factors, such as the team and the distance divided by 100, to facilitate interpretation, and the batch. Then, we ran another model only with climatic factors, such as wind intensity, temperature, humidity, and precipitation. Prevalence ratios (PRs) and the corresponding 95% confidence intervals (CIs) were estimated for each variable.

All the analyses were performed with R version 4.2.1. and RStudio version 2022.12.0+353 [34,35]. The Kernel Density Estimation interpolations (heat maps) were developed in QGIS, version 3.16.9-Hannover, with a background map from OpenStreetMap (CC-BY-SA).

## 3. Results

### 3.1. Recapture Rate

A total of 84,000 marked sterile male mosquitoes were released in three batches during three consecutive weeks. Of these, a total of 20,712 (24.7%) sterile male mosquitoes did not fly out from the release cups after 60 min. The mean recapture rate across the study was 0.8%, considering that 528 marked males were recaptured. Climatic conditions were similar during all releases conducted: the mean daily temperature varied between 21.0 and 22.4 °C, the humidity was between 70 and 88%, and the wind intensity was weak. The gathered information from the MRR study is summarized in Table 1, presenting the number of released marked sterile mosquitoes per fluorescent dusk color/release, and the release point, as well as the estimated parameters.

Overall, the proportion of marked mosquitoes recaptured per color varied between 0.32% and 1.23% (Figure 2). Although the recapture rate was low throughout the whole experiment, we observed statistically significant differences comparing the batches marked with the color green and batches marked with pink, red, and yellow (Appendix A).

### 3.2. Dispersal and Spatial Distribution

Marked sterile males had an average MDT of 88.7 m, with a minimum MDT of 58.7 m and a maximum MDT of 160.8 m from the release points (Table 1). The maximum distance travelled by sterile males under local conditions was 299.0 m. FR50 and FR90 were estimated per color (Table 1) and bootstrapped to obtain an estimation for the sterile marked males. The bootstrapped estimates for FR50 and FR90 were 42.8 m (95% BCa CI: 27.6; 73.6), and 170.6 m (95% BCa CI: 143.7; 228.9), respectively. Figure 3 shows the spatial distribution of the released males. Overall, the central zone of the study area showed the highest recapture rates and frequencies, i.e., mosquitoes tended to stay closer to the release point, and visually, there was no indication that the primary wind direction had any influence on mosquito dispersal. However, some colors, particularly pink, showed a distinct distribution pattern.

### 3.3. Wild Population Size

The population size was estimated for each performed release (Table 1) and bootstrapped; the estimates for the wild male population in the study area was 11,970 mosquitoes or 443.33 males/ha. Considering the wild male–female ratio (i.e., 0.98), the estimated population size was approximately 11,730 wild females, and 23,700 wild fertile mosquitoes (both males and females), corresponding to 877.78 mosquitoes/ha.

### 3.4. Sterile-to-Wild Male Ratio

The sterile-to-wild male ratio showed the dynamic between the two male types. Overall, this parameter was low over the entire experiment, with an average value of 0.86 sterile-to-wild ratio. Table 1 and Figure 4 show the ratio for each color, and the dynamics of the ratios over time, respectively. Results relating to the sterile-to-wild male ratio clearly show how this parameter decreases over time following the release, increasing only when a subsequent release was performed.

### 3.5. Survival

The PDS and ALE were 60% and 1.95 for blue, 60.7% and 2.01 for red, 61.8% and 2.08 for pink, 63.9% and 2.23 for yellow, 102.7% and −37.55 for orange, and 104.7% and −21.84 for green color release batches. The orange and green values corresponded to negative or biologically non-sense values and do not reflect the field.

### 3.6. Influence of Climatic Factors on Capture

During HLCs, the temperature was, on average, 22 °C; humidity was, on average, 81%, mostly without precipitation; and the wind direction was mostly southerly and weak (Appendix A). The capture rate was significantly impacted by weather conditions, with the lowest rates observed during the second release when different climatic conditions such as a stronger wind intensity, lower temperature, precipitation, and high humidity were recorded (Appendix A). We found that the proportion of recaptured mosquitoes was 23% lower when the wind intensity was moderate compared with a weak wind intensity (PRa: 0.77, 95% CI: 0.61; 0.98). There was no statistically significant association between the number of recaptured mosquitoes and temperature, precipitation, and humidity. However, the number of mosquitoes recaptured was 2% lower for each percentage increase in humidity above the mean humidity during the captures (PRa: 0.98, 95% CI: 0.96; 1.00) (Table 2).

## 4. Discussion

The successful application and subsequent implementation of the SIT is context dependent regarding local environmental conditions. At this development stage of the SIT applied to the control of *Aedes* mosquitoes, it is extremely important to perform field trials, such as the one presented in this study, to develop and optimize an efficient operational technology to be applied successfully, as we have had in other examples [19,20,21,22,23].

Mark–release–recapture (MRR) studies are valuable for understanding insect population field parameters [36,37,38]. These studies involve marking insects, releasing them, and then recapturing them periodically [27]. Analyzing recapture rates makes it possible to understand the bio-ecological features of the target population and how the laboratory-produced sterile males may perform in the natural environment.

To plan the SIT interventions effectively, understanding the traits of sterile males and accurately quantifying wild population density, dispersion patterns, flight distances, and survival rates are essential. In our study, we resorted to MRR studies to assess these parameters, which will be crucial in planning the SIT pilot field trials in Portugal to suppress *Ae. albopictus* populations. Moreover, we considered that meteorological conditions (temperature, relative humidity, and wind speed) influence the results by affecting mosquito abundance, survival, and dispersal [38,39]. So, in our study, we focused on how climatic factors affect the capture/recapture of mosquitoes and influence the estimated parameters.

It is important to highlight that in this study, the sterile males were mass reared at the CAA laboratory in Bologna, Italy, and shipped from there to the study site by adopting a transporting procedure developed by CAA, as described under Section 2.3. This approach followed similar attempts in Europe and the successful transport of sterile males is a critical factor for the implementation of effective SIT programs, since these are dependent on the quality of the sterile males, which in turn affect the overall results [28].

### 4.1. Recapture Rate

In our study, the recapture rate reached 0.8% of the total release. A similar MRR study in Switzerland, in two different localities, resulted in rates of 9.3% and 2.1%, which points to a low recapture rate, although this was higher than ours [32]. In an MRR study performed in Albania, the reported recapture rate of 3.93% was also higher [38]. However, our results showed higher recapture rates considering other studies on MRR trials [40,41,42,43]. The variation in these results is related to the quality of the sterile males and the influence of climatic conditions on the recapture periods. Such low recapture rates can be connected with stress during transportation and the collection method. For instance the use of BGS traps can have a positive impact on recapture rates [38]. Additionally, our results may have been affected by the longer air transportation suffered by the sterile adults in comparison with any other MRR recapture study performed with *Ae. albopictus.* The improvement in shipping conditions will be important and could enhance the performance of males. Moreover, the problem connected with long distance transportation in term of quality and the associated logistics strongly suggest the need for the local capacity for mass production of sterile males in view of a national enlargement of the control program integrating the SIT.

### 4.2. Dispersal and Spatial Distribution

Factors such as wind, humidity, temperature, and rainfall are known to affect the flight range of mosquitoes [44]. Moreover, it has already been proven that extreme climatic conditions negatively affect the survival and dispersal of *Ae. albopictus* mosquitoes. In a comparison between MRR trials in which two of them were carried out under optimal climatic conditions and a third was affected by heavy rainfall and a severe drop in temperature, the latter showed a much lower recapture rate. Additionally, delayed oviposition dynamics between released females and females kept in sheltered cages, and a lower-distance dispersal were observed in another study [45]. In our study, since the recapture rates were low, it was not possible to identify hot spots resulting from environmental conditions.

Additionally, the wind was identified as an important factor influencing the dispersal of mosquitoes, and mosquito-borne diseases have been the subject of several studies [46]. In a MRR experiment, Marcantonio and colleagues (2019) showed that the probability of recapturing marked *Ae. aegypti* (Linnaeus, 1762) was affected by wind speed [39]. Moreover, wind speed and wind direction can affect the dispersal range of *Ae. albopictus* [47]. In this study, the mean wind direction did not directly affect the sterile male mosquito distribution. However, the moderate wind intensity could explain why the mosquitoes did not disperse since they tend not to move under these conditions. As in previous studies, our results suggest that environmental data can help to understand specific patterns besides the mosquito data collection. Even though the marked mosquitoes had a mean distance travel of around 59 to 161 m, these distances are similar to the distances found in other studies, such as the study in Albania with 77 up to 99 m from the release point [38] and in the USA with 202 m from the release point [48].

### 4.3. Population Size

This information is essential to dimensioning the sterile male production and release doses aiming to suppress the wild population. In the target area, we estimated the mosquito population was around 11,970 males and, including females, the total was around 23,700 wild mosquitoes, corresponding to 877 mosquitoes/ha. This value is in the range of other studies, in which they found 134 mosquitoes/ha and 767 mosquitoes/ha from two different areas [32]. But, this estimation might be an overestimation due to the mortality that the males may experience during the release phase, which cannot be estimated at this stage.

### 4.4. Sterile-to-Wild Ratio

Understanding the size of the wild male population in a given area is essential for calculating the necessary amount of sterile males to be released. This information helps to achieve the optimal sterile-to-wild male ratio, which is crucial for effective population control. The expected sterile-to-wild ratio to reach suppression levels should fall between 10 to 100 [19]. Our study had a mean sterile-to-wild ratio of 0.86, while an MRR study in Albania had an overall sterile-to-wild ratio of 0.45 [38]. Longer SIT pilot trials showed mean values around 44.82, and another study ranged from 22.95 to 350.47 [22]. In both cases, the continuity of the SIT program can use the sterile-to-wild value to adjust the release numbers to achieve desired suppression levels. It is noteworthy that studies are showing that irradiation treatment does not seem to considerably alter the parameters of flight (flight speed or distance flown) that may influence competitiveness with wild males [49].

This ratio is a crucial parameter to the success of the SIT, which provides the release density of sterile males needed in that season. The proportion of sterile/wild males must be enough to suppress the wild population. And indeed, the MRR studies provide us with this essential information to properly plan the release strategy. Also, our results make it clear that the suppression cannot be initiated during this season (October), when natural population abundance is high, as a larger quantity of sterile males would be needed to achieve the desired suppression rate.

### 4.5. Survival

This relevant parameter provide us with information on sterile male quality in relation to local environmental conditions. The handling and transportation process used in this study may have affected the male mosquitos’ survival, as was mentioned previously.

In our study, the probability of daily survival (PDS) was 60.0–63.9% and the average life expectancy (ALE) was 1.95–2.23 (Table 1). These results are lower than those reported from the MRR study with *Ae. albopictus* in Albania [38], but in line with other studies in which irradiated mosquitoes were released from the ground [50,51,52]. The survival rate of sterile mosquitoes might have been impacted by the recapture method and the climatic conditions, interfering with calculating the probability of daily survival (PDS), and thus affecting the average life expectancy (ALE). The PDS is essential to determine the frequency of release of sterile males in SIT programs, taking into account other parameters, namely the dispersal and abundance of the wild population. Our results highlight that the release of sterile male should be performed at least three times per week so the study area can be covered by sterile males for the whole week.

### 4.6. Influence of Climatic Factors on Capture

The recapture rate was significantly impacted by weather conditions, with the lowest rates observed during the second release when different climatic conditions such as moderate wind intensity, lower temperature, precipitation, and high humidity were recorded (Appendix A). The mean wind direction did not directly affect the sterile male mosquito distribution. However, the moderate wind intensity affected the recapture rate inferring reduced mosquito dispersion, since mosquitoes tend not to fly in these conditions.

The mean temperature during recaptures was 22 °C and the median was 21.80 °C (IQR: 21.10–22.50 °C). This small variation explains the lack of association between temperature and the number of mosquitoes captured. The same can apply to the humidity parameter.

This MRR study was conducted in October. It is essential to recognize that the performance of released sterile males can fluctuate during the mosquito season due to environmental variations [53]. Factors such as dispersal, survival, and the wild population size play a crucial role. Consequently, release strategies should consider both population quality and size at different time points during the season. Nonetheless, to enhance our understanding of population dynamics during the SIT pilot trials, it is essential to combine seasonal baseline data collection with the MRR study results following the phase conditional approach [18], to enable accurate estimations over time.

## 5. Conclusions

We analyzed the dispersal and spatial distribution of the sterile male *Ae. albopictus* mosquitoes, the wild population size, the ratio of sterile-to-wild male mosquitoes, and the survival rate of the released males, along with the influence of weather factors on these parameters. This knowledge, with the baseline data on mosquito seasonality and distribution, will contribute to a more efficient application of the SIT in an integrated vector management program.

Our results could have been affected by the air transportation logistics of the irradiated sterile mosquitoes, which may have impacted their quality. However, considering the mortality observed after shipping, the results obtained were extremely encouraging. The improvement in shipping conditions will be important and could enhance the performance of the radio-sterilized males. This highlights the need for local capacity for mass production and sterilization in view of a national enlargement of the control program integrating the SIT.

It is crucial to plan MRRs carefully, considering weather conditions during their execution to avoid underestimating the number of sterile males needed for the SIT and other parameters such as dispersion and survival rate compromising the effectiveness of this control method.

In the absence of a mosquito vector control strategy, the invasive *Ae. albopictus* will quickly disseminate and establish in other Portuguese regions, increasing the risk of mosquito-borne disease outbreaks. Recent data shows that the initial *Ae. albopictus* population is spreading to neighboring areas, reaching wider areas, and becoming established in new counties surrounding the first documented location (unpublished data). Benefiting the prevention rather than mitigation, the early implementation of an integrated vector control intervention that reduces the abundance of mosquito populations and confines their geographical distribution is a crucial strategy that must engage several activities, but most importantly, the public and environmental health policies.

## Figures and Tables

**Figure 1 insects-15-00685-f001:**
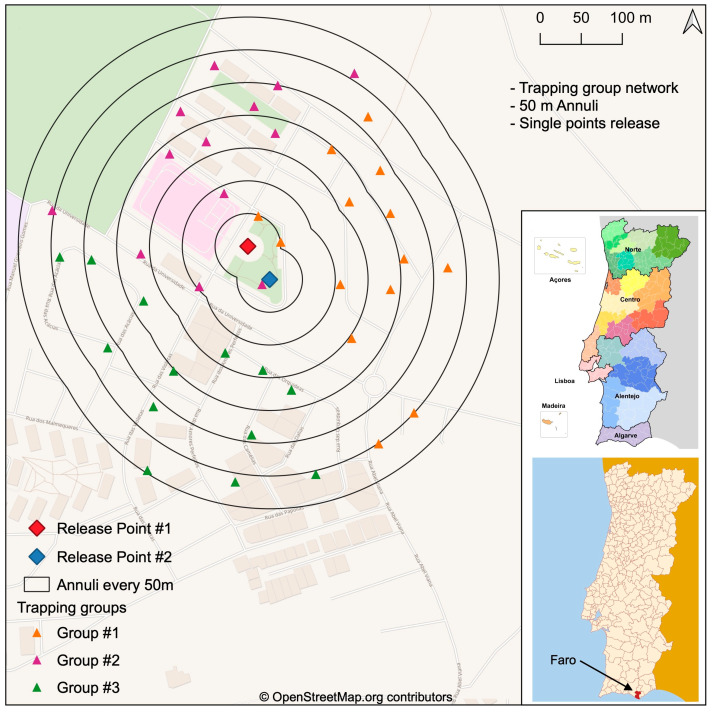
Target area of Gambelas, Faro municipality, Algarve, Portugal. Two release points were set. From those points, sequential annuli with 50 m distance of up to 350 m were estimated, resulting in an area of approximately 27 ha. Mosquito recaptures were performed by human landing collection (HLC), and these predefined 40 HLC stations were initially divided among three collection groups of technicians to facilitate synchronized collection.

**Figure 2 insects-15-00685-f002:**
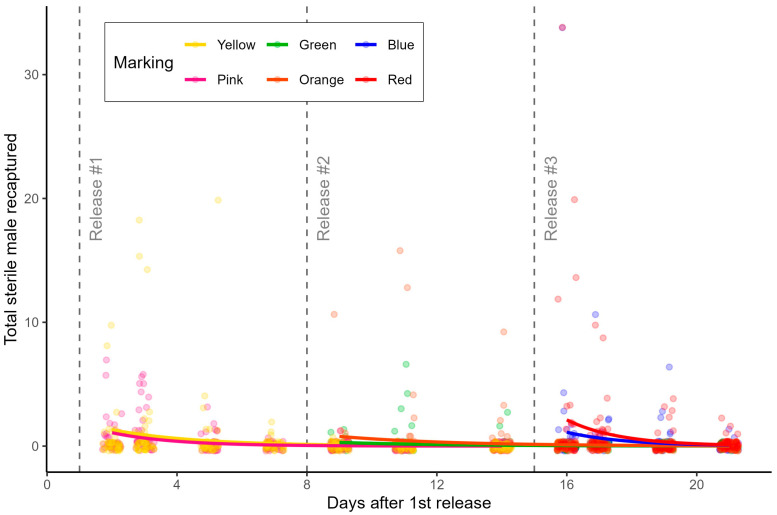
Total recapture of marked sterile males from different release color batches in Faro, Portugal.

**Figure 3 insects-15-00685-f003:**
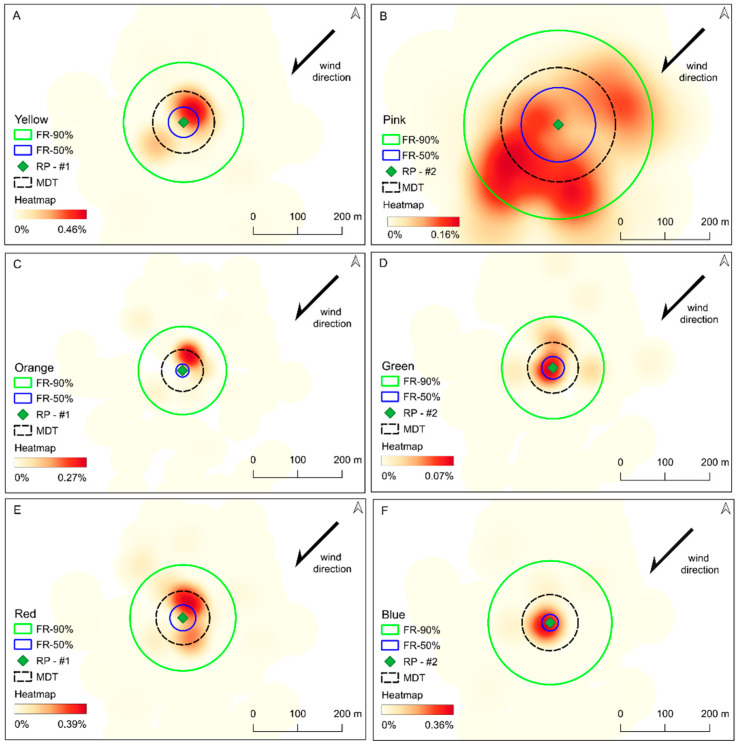
Spatial distribution of marked sterile *Ae. albopictus* in three different batches with two replicates each in reference to its respective mean distance travelled (MDT) and 50% and 90% flight range (FR-50% and FR-90%). (**A**–**F**) represent the spatial distribution of the corresponding release colors: yellow, pink, orange, green, red, and blue. (**A**,**B**) corresponds to the first release; (**C**,**D**) second release; and (**E**,**F**) third release. The predominant wind direction during the collection period is indicated by the arrow according to each color. The MDT, FR-50%, and FR-90% corresponded to the respective color’s release point.

**Figure 4 insects-15-00685-f004:**
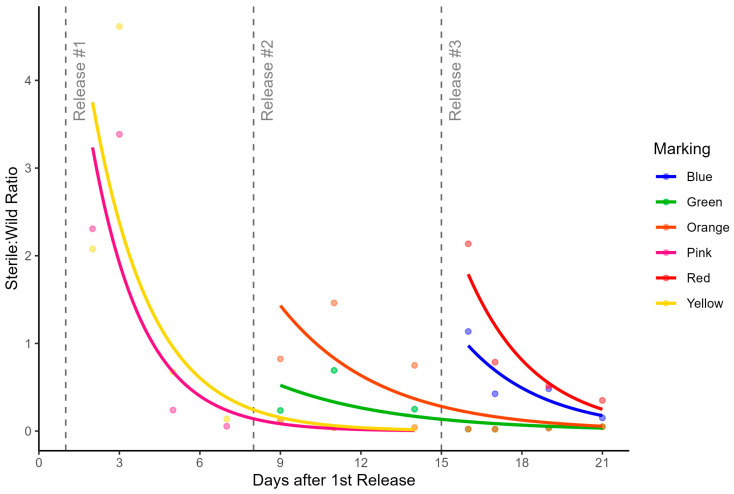
The proportion ratio of marked sterile males for each wild male captured during the MRR using distinct color release batches in Faro, Portugal.

**Table 1 insects-15-00685-t001:** Summary table of the MRR parameters obtained in the analyses for each color release of the three *Ae. albopictus* batches.

Color	Batch #1		Batch #2		Batch #3	
	Pink	Yellow	Green	Orange	Blue	Red
Release date	11 October 2022	11 October 2022	18 October 2022	18 October 2022	25 October 2022	25 October 2022
Release site	2	1	2	1	2	1
Released, n	14,000	14,000	14,000	14,000	14,000	14,000
Mortality, n (%)	5126 (36.61%)	1186 (8.47%)	6286 (44.9%)	3346 (23.9%)	2927 (20.91%)	1841 (13.15%)
Recapture, n (%)	84 (0.95%)	120 (0.94%)	25 (0.32%)	67 (0.63%)	83 (0.75%)	149 (1.23%)
PDS (95% CI)	61.8% (45.4%; 83.9%)	63.9% (43.5%; 93.7%)	-	-	60% (40.1%; 89.6%)	60.7% (46.1%; 80.1%)
ALE	2.08	2.23	-	-	1.95	2.01
MDT	160.76 (15.98)	87.01 (12.72)	71.42 (13.49)	58.67 (8.54)	78.75 (4.73)	75.53 (10.53)
FR50 (95% CI)	104.57 (76.06; 143.78)	42.71 (24.83; 73.45)	31.8 (15.3; 66.08)	18.29 (9.22; 36.28)	23.38 (18.08; 30.23)	36.26 (23.39; 56.22)
FR90 (95% CI)	265.33 (167.6; 420.03)	168.41 (116.9; 242.63)	143.54 (97.88; 210.5)	123.83 (94.77; 161.79)	173.7 (152.84; 197.41)	148.68 (115.09; 192.07)
Population	2267.81	3776.77	43,068.56	13,466.51	5825.70	3414.77
Sterile-to-wild ratio	0.78	1.15	0.39	1.05	0.61	1.10

PDS—probability of daily survival, ALE—average life expectancy, MDT—mean distance travelled, FR—flight range. Missing values for PDS and ALE corresponded to negative or biologically non-sense values.

**Table 2 insects-15-00685-t002:** Association between the number of captured mosquitoes and climatic conditions (PR: prevalence ratio; CI: confidence interval; significant values are displayed in bold).

	PR	95% CI	*p*-Value
Wind intensity (ref. weak)	**0.77**	**0.61, 0.98**	**0.03**
Temperature (centered at the mean)	1.03	0.83, 1.28	0.78
Humidity (centered at the mean)	**0.98**	**0.96, 1.00**	**0.09**
Precipitation (ref. none)	1.07	0.60, 2.03	0.82

## Data Availability

The data presented in this study are available on request from the corresponding author.

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
