# Peer review of "Mark–Release–Recapture Trial with Aedes albopictus (Diptera, Culicidae) Irradiated Males: Population Parameters and Climatic Factors"

_insects, 2024, doi:10.3390/insects15090685_

Round 1
Reviewer 1 Report
Comments and Suggestions for Authors
My comments are in attached document

Many grammar mistakes and spelling mistakes. See my recommendations in the attached document.
Author Response
Dear reviewer
Thank you very much for your revision report, which greatly improved the quality of the manuscript.
Please consider the attached word doc, in which, highlighted in blue, you can see all the grammar corrections that you have suggested and were fully accepted. In the doc you may also found the answers to your questions and comments that I list below point by point:
Comment 1: L65 To my knowledge, Ae. albopictus does not transmit parasites.
Response 1: Aedes albopictus is mostly known as a virus carrier but it is also a natural vector of Dirofilaria immitis in Italy: Vet Parasitol. 2003 Dec 30;118(3-4):195-202. doi: 10.1016/j.vetpar.2003.10.011.
Comment 2: L129 How was this estimated? Is this a different study?
Response 2: The wild population was estimated using the Fisher-Ford index, explained in the methodology. The references are:
- Cianci_2013_Estimating Mosquito Population Size From Mark–Release–Recapture Data in the Literature review folder of the MRR
- Carvalho_2022 - Captiva MRR
Comment 3: L135 It could important to mention that when using SIT the ratio of sterile to wild mosquitoes can go from 5-10 to 1.
Response 3: These numbers are mentioned in the discussion lines 431-436.
Comment 4: L162 What does mean? Optimum temperature, precipitation based on previous information
Response 4: We agree that the term is subjective. We have removed these words.
Comment 5: L191 Here is a contradiction. Releases were at one-week intervals per 3 weeks. Rephrase this sentence please. For example, Sterile mosquitoes were release twice a week for three consecutive weeks.
Response 5: The mosquitoes were released once a week, at the same time but in two nearby release points depicted on Figure 1.
Comment 6: L291 It will be worthy to mention that recaptures were higher in site 1 in all cases. Explain why this is the case?
Response 6: The release points were very close to each other and we have no data to support that recaptures in release point 1 were higher than in release point 2. Although we found differences between batches, considering the mean distance travelled (89m) we cannot assume that recaptures were significantly higher in release point 1.
Comment 7: L396 It will be interesting to assess what factors are different between site 1 and site 2 since in site 1 recaptures were higher (> 60%) compared to site. I would highly recommend that site should be included in the models? Or you can have model with site and without sites to assess the differences.
Response 7: As mentioned above, the release points were very close to each other and we have no data to support that recaptures in release point 1 were higher than in release point 2. Although we found differences between batches, considering the mean distance travelled (89m) we cannot assume that recaptures were significantly higher in release point 1.
Comment 8: L427 In this case is better to overestimate than underestimate. Also, included a paragraph highlighting the ratio of sterile to wild mosquito for wild mosquito population reduction. Also, include information on how trapping is important to monitor population reduction. For example, number of traps needed when populations are very small (Power analysis).
Response 8: We discussed the ratio of sterile to wild mosquito in lines 433-438. In this study, we did not use traps, which can be a limitation and is acknowledge in line 387-389. That is why we did not discuss this issue at length.
Comment 9: L482 I don’t if this is relevant for a conclusion.
Response 9: We agree and have deleted the sentence.

Reviewer 2 Report
Comments and Suggestions for Authors
1. All species names should be written out in full and the describing author should be provided the first time they are mentioned; subsequently, abbreviations should be used throughout the text. Aedes albopictus (Skuse, 1894). All genus and species names should be written in italics.
2. I suggest that the references should be organized according to the journal spelling rules (Publication dates bold/nonbold, Punctations after Journal names-Full-itatic or not or Abbreviations, Live type names, species names should be italic etc.)
3. Line 48; 84,000–sterile should be revised as 84,000 sterile
4. Line 52; 443.33 males/ ha shoud be revised as 443.33 males/ha. Line 315 also.
5. Lines 57-59; Keywords make more sense when listed in alphabetical order.
6. Lines 65-66; Diseases should be listed with their initial letter capitalized, such as Dengue, Zika, Yellow Fever, Japanese Encephalitis, and Chikungunya."
7. Lines 171-173; It is mentioned that male pupae are exposed to radiation in the sterile insect technique. They should briefly explain within the text how they distinguish between male and female individuals in the pupal stage.
8. Lines 171-184; The time from being taken from the laboratory to being released into nature is specified as 28 hours, but what is the approximate age of the sterile male adults when released into nature? Is it one or two days?
9. Was an ultra-low volume or residual application of any insecticide made in the area where sterile insects were released between October 11 and 31, 2022? This situation should be mentioned in the text.
10. In this study, the Human Landing Collection (HLC) method was used, and 40 people took part in the tests. Was Ethics Committee permission obtained for the research?
11. Doesn't the microclimate and the spatial arrangement of hosts affect the direction and speed of mosquito dispersal?
12. Lines 377-392; Could geographical barriers and vegetation also play a role in the recapture of more mosquitoes in other studies than in yours? Please give more information about vejetation type and barries of the study area.
13. Line 418 Release doses refer to the amount of a substance (e.g., insecticide, medication) released at a given time. Therefore ‘release doses’ or ‘release rates’ which is suitable?
Author Response
Comment 1: All species names should be written out in full and the describing author should be provided the first time they are mentioned; subsequently, abbreviations should be used throughout the text. Aedes albopictus (Skuse, 1894). All genus and species names should be written in italics.
Response 1: Thank you for pointing this out. Describing author provided the first time Aedes albopictus (L-62) and Aedes aegypti (L-408) are mentioned. The authors confirm that all genus and species names are in italics.
Comment 2: I suggest that the references should be organized according to the journal spelling rules (Publication dates bold/nonbold, Punctations after Journal names-Full-itatic or not or Abbreviations, Live type names, species names should be italic etc.)
Response 2: The references section was reviewed and organized accordingly.
Comment 3: 84,000–sterile should be revised as 84,000 sterile
Response 3: Done.
Comment 4: Line 52; 443.33 males/ ha shoud be revised as 443.33 males/ha. Line 315 also.
Response 4: Thank you. Done.
Comment 5: Lines 57-59; Keywords make more sense when listed in alphabetical order.
Response 5: Agree. We have modified accordingly.
Comment 6: Lines 65-66; Diseases should be listed with their initial letter capitalized, such as Dengue, Zika, Yellow Fever, Japanese Encephalitis, and Chikungunya."
Response 6: We appreciate the suggestion, but we usually follow the Associated Press guidelines, where the names of the diseases are only capitalized when they are known by the name of a person or geographical areas. In this way, dengue and chik must not be capitalized, but Zika must be.
Comment 7: Lines 171-173; It is mentioned that male pupae are exposed to radiation in the sterile insect technique. They should briefly explain within the text how they distinguish between male and female individuals in the pupal stage.
Response 7: Male and female mosquito pupae may be distinguish and separeted by size. Females are larger than males. Referred in text body.
"One of the most common approaches for mosquito sex separation is size-based sorting, where female pupae are typically larger than males" Efficiency assessment of a novel automatic mosquito pupae sex separation system in support of area-wide male-based release strategies (nature.com)
Comment 8: Lines 171-184; The time from being taken from the laboratory to being released into nature is specified as 28 hours, but what is the approximate age of the sterile male adults when released into nature? Is it one or two days?
Response 8: Sterile male mosquitoes produced in this study had 32-48 hours age when released into nature.
Comment 9: Was an ultra-low volume or residual application of any insecticide made in the area where sterile insects were released between October 11 and 31, 2022? This situation should be mentioned in the text.
Response 9: No insecticide was made in the target or control area during the period of this study, not even in the 6 moth previous period. Now referred in body text L-145-146.
Comment 10: In this study, the Human Landing Collection (HLC) method was used, and 40 people took part in the tests. Was Ethics Committee permission obtained for the research?
Response 10: The research project was approved by the Ethics Commmitee for Health (CES-INSA) and CES-INSA gave its favorable opinion.
Comment 11: Doesn't the microclimate and the spatial arrangement of hosts affect the direction and speed of mosquito dispersal?
Response 11: We used data from a meteorological station situated near the target area. Thus, we were unable to identify possible microclimate variations in our study. Considering the effect of the wind on mosquito capture, a microclimate could affect the dispersal of the mosquitos, thus further influencing our results.
Comment 12: Lines 377-392; Could geographical barriers and vegetation also play a role in the recapture of more mosquitoes in other studies than in yours? Please give more information about vejetation type and barries of the study area.
Response 12: Site-specific geographical barriers and vegetation also play a role. The surrounding area of the neighbourhood is mainly composed by pine tree forests, as it is depicted in the text. The vegetation in the private gardens is very diverse, with some trees, flowers or bushes, and very often (almost always) not easily seen from the outside. In this way, we cannot conjecture too much about the influence of the vegetation in this study.
Comment 13: Line 418 Release doses refer to the amount of a substance (e.g., insecticide, medication) released at a given time. Therefore ‘release doses’ or ‘release rates’ which is suitable?
Response 13: We understand, however in this case we consider correct to write release doses since in this paragraph we are not refering to a rate, but to an absolute number of mosquitoes per dose needed to supress the wild population.

Reviewer 3 Report
Comments and Suggestions for Authors
This paper has significant contribution to the overall efforts to combat Ae.A., especially for the estimation of the size of the wild population and dispersal of Ae.A.
Introduction is well prepared, only one suggestion is that would be interesting for the readers to breefly explain Ae.A. biological cycle and level of the Mass rearing of Ae.A: bisexual strain production, duration, sorting etc. with 2-3 sentences.
MM section is well explained and understandable. In the MM section Mosquito production, would be nice and interesting for the readers to explain treatment of adults with fluorescent dye instead just IAEA reference.
Results are well presented (recapture rate, population size, s:w ratio,.
Results regarding Influence of climatic factor 341-355, suggest that Title of the MS is not adequate, since there is no effect of the climatic conditions. This paper however present male performances in the field and so far the title can be adjusted a bit.
Discussion 383-392: looks like parameters of the quality of sterile males affect overall results not only recapture rate. I suggest that this paragraph can be moved above in the Discussion after line 375
Conclusions: 478: state what is the 2nd phase
Author Response
Comment 1: Introduction is well prepared, only one suggestion is that would be interesting for the readers to breefly explain Ae.A. biological cycle and level of the Mass rearing of Ae.A: bisexual strain production, duration, sorting etc. with 2-3 sentences.
Response 1: Your suggestion is much apreciated, but we believe that the present text, in addition to the references, is enough.
Comment 2: MM section is well explained and understandable. In the MM section Mosquito production, would be nice and interesting for the readers to explain treatment of adults with fluorescent dye instead just IAEA reference.
Response 2: Thank you for the suggestion. We considered for this procedure (1000-2000 inmobilized by low temperature mosquitoes in a jar with fluorescent pownder the reference with the technical details is enough.
Comment 3: Results regarding Influence of climatic factor 341-355, suggest that Title of the MS is not adequate, since there is no effect of the climatic conditions. This paper however present male performances in the field and so far the title can be adjusted a bit.
Response 3: Thank you for this suggestion. The title of the manuscript was adjusted to: Mark-Release-Recapture trial with Aedes albopictus (Diptera, Culicidae) irradiated males: population parameters and climatic factors
Comment 4: Discussion 383-392: looks like parameters of the quality of sterile males affect overall results not only recapture rate. I suggest that this paragraph can be moved above in the Discussion after line 375.
Response 4: Thank you vry much for pointing this out. We choose to kept the text and not move the paragraph, but added the following sentence in the discussion L375-379: This approach followed similar attempts in Europe and the successful transport of sterile males is a critical factor for the implementation of effective SIT programs, since these are dependent on the quality of the sterile males, which in turn affect overall results.
Comment 5: Conclusions: 478: state what is the 2nd phase
Response 5: The second phase is the SIT pilot trial itself. To avoid doubts we changed second phase by a later.

Round 2
Reviewer 1 Report
Comments and Suggestions for Authors
In line 65 you are talking about dirofilariasis (correct?) It will be better to mention that A. albopictus also transmit nematodes of the genus Dirofilaria to carnivore hosts. Some Dirofilaria species are are zoonotic.
It you leave like viruses and parasites it seems that transmit multiple parasites.
I could not find any response to one of my main questions. What factors explains the difference between site 1 and site 2 in recapture of mosquitoes?
Also, why site was not included in the model if you have differences of more than 50% in recaptures between sites; in batch 2 more than 100%?
Comments on the Quality of English Language
NA
Author Response
Comment 1: In line 65 you are talking about dirofilariasis (correct?) It will be better to mention that A. albopictus also transmit nematodes of the genus Dirofilaria to carnivore hosts. Some Dirofilaria species are are zoonotic.
It you leave like viruses and parasites it seems that transmit multiple parasites.
Response 1: Thank you for reinforcing this point. We agree to change. Please check the sentence:
L65 "The mosquito Aedes albopictus (Skuse, 1884), also known as the Asian Tiger Mosquito, is a competent vector species of a wide range of viruses and nematodes of the genus Dirofilaria spp., which raises a big concern in veterinary and public health [3]."
Comment 2: I could not find any response to one of my main questions. What factors explains the difference between site 1 and site 2 in recapture of mosquitoes?
Response 2: The two release points are bellow 50 meters far away from each other in the same urban park and we looked from the beginning for similar release conditions to release both marked samples in each batch. However, during the study we realized that Site 2 had always higher mortality than Site 1 (table 1), meaning not only dead mosquitoes, but mosquitoes that were alive in the sample and did not fly. So to calculate mortality we count all the mosquitoes that were dead, which we presumed to be the same for both release sites, based on the transportation conditions were the same, and mosquitoes that were alive but did not flew out the release cup or the blank towel in the ground (ground releases). After the study, we realized that the ground temperature could had an impact on the viability of the mosquitoes to flew out the release cups (so in the mortality) and in the process of finding a food source on the very first time essential to survive. After measuring the ground temperature we found a diference of 2-5 degrees from site 1 (higher ground temperature) to site 2. So, this could be the factor that affected the release profiles of sites 1 and 2, and therefore making the mosquitoes released from site 2 less viable and fit to survive, and finally affecting the recapture rate between the mosquitoes released in those sites.
Comment 3: Also, why site was not included in the model if you have differences of more than 50% in recaptures between sites; in batch 2 more than 100%?
Response 3: We apologise for the confusion and thank the Reviewer for the opportunity to clarify what was done. The models did not include the site as the outcome corresponds to the mosquito capture and not only recapture. This is mentioned in the methodology. Climatic conditions can affect both wild and sterile mosquitoes, thus we modelled the number of captured mosquitoes and not recaptured mosquitoes. Therefore, adding the site to the models will be misleading, as wild mosquitos don't have an associated site.
